# Transcriptome Analysis Reveals That SREBP Modulates a Large Repertoire of Genes Involved in Key Cellular Functions in *Penaeus vannamei*, although the Majority of the Dysregulated Genes Are Unannotated

**DOI:** 10.3390/genes13112057

**Published:** 2022-11-07

**Authors:** Xiaoyu Zheng, Zishu Huang, Zhuoyan Liu, Zhihong Zheng, Yueling Zhang, Jude Juventus Aweya

**Affiliations:** 1College of Ocean Food and Biological Engineering, Fujian Provincial Key Laboratory of Food Microbiology and Enzyme Engineering, Jimei University, Xiamen 361021, China; 2Institute of Marine Sciences and Guangdong Provincial Key Laboratory of Marine Biotechnology, Shantou University, Shantou 515063, China; 3Southern Marine Science and Engineering Guangdong Laboratory, Guangzhou 511458, China

**Keywords:** SREBPs, *Penaeus vannamei*, transcriptome, metabolism, immune response, cellular signaling

## Abstract

Sterol regulatory element-binding proteins (SREBPs) play vital roles in fatty acid metabolism and other metabolic processes in mammals. However, in penaeid shrimp, the repertoire of genes modulated by SREBP is unknown. Here, RNA interference-mediated knockdown followed by transcriptome sequencing on the Illumina Novaseq 6000 platform was used to explore the genes modulated by SREBP in *Penaeus vannamei* hepatopancreas. A total of 706 differentially expressed genes (DEGs) were identified, out of which 282 were upregulated and 424 downregulated. Although gene ontology (GO) and Kyoto Encyclopedia of Genes and Genomes (KEGG) analyses revealed that most of the downregulated DEGs were involved in physiological processes related to immunity, metabolism, and cellular signaling pathways, many of the dysregulated genes have uncharacterized functions. While most of the dysregulated genes were annotated in metabolic processes, such as carbohydrate metabolism, lipid metabolism, signal transduction, and immune system, a large number (42.21%) are uncharacterized. Collectively, our current data revealed that SREBP modulates many genes involved in crucial physiological processes, such as energy metabolism, immune response, and cellular signaling pathways, as well as numerous genes with unannotated functions, in penaeid shrimp. These findings indicated that our knowledge of the repertoire of genes modulated by SREBP in shrimp lags behind that of mammals, probably due to limited research or because the complete genome of *P. vannamei* has just been sequenced.

## 1. Introduction

Sterol regulatory element-binding proteins (SREBPs) are transcription factors that maintain cellular lipid metabolism by controlling the synthesis of fatty acids, triglycerides, and cholesterol [1]. As well as their role in lipid metabolism and as regulators of cellular homeostasis [1,2], mammalian SREBPs are involved in immune response [3], metabolic reprogramming to enhance glycolysis and oxidative phosphorylation [4], reprogramming of fatty acids metabolism via Toll-like receptor 4 (TLR4) signaling to resolve inflammatory processes in macrophages [5], and regulation of lipid synthesis in macrophages to stimulate TLR4-dependent phagocytosis [6]. Similarly, SREBPs modulate the expression of many genes in various tissues not directly related to fatty acid metabolism and lipogenesis, including the regulation of other membrane-bound transcription factors such as ATF6 and CREB3 [7,8], energy balance and glucose homeostasis [9], increase in intracellular triacylglycerols (TAG) levels [10], and insulin resistance [11].

Mammals express three SREBP family members, i.e., SREBP-1a, SREBP-1c, and SREBP-2 [12], encoded by *Srebf1* or *Srebf2* genes [13]. SREBP-1 is more specific to fatty acid synthesis [14], while SREBP-2 is the master regulator of cholesterol synthesis and lipogenesis [15]. Two SREBP homologs (Srebp1 and Srebp2) have been reported in fish [16] and implicated in long-chain polyunsaturated fatty acids (LC-PUFA) biosynthesis and regulation of fatty acid metabolism-related genes [17]. On the other hand, a single SREBP gene has been identified in invertebrates thus far [18], including one SREBP homolog in the noble scallop (*Chlamys nobilis*) [19], Asian marine razor-clam (*Sinonovacula constricta*) [20], mud crab (*Scylla paramamosain*) [21], and pacific white shrimp (*Penaeus vannamei*) [22]. These invertebrate SREBP homologs have been implicated in the regulation of various aspects of lipid metabolism, molting, and ecdysteroidogenesis [22], and immune response [23].

Under normal physiological conditions, SREBPs, which are initially synthesized as precursors bound to the endoplasmic reticulum (ER), are transported to the Golgi apparatus, where two-step sequential proteolytic cleavage activation releases the active form that enters the nucleus to regulate the transcription of target genes [24]. The processing of SREBPs is generally controlled by cellular sterol content [25]. In mammals, the various SREBP isoforms can modulate the expression of many genes not necessarily involved in lipid metabolism. For instance, the attenuation of SREBP-1 expression could result in different physiological changes, including decrease in the mRNA/protein levels of glucagon-like peptide 2 receptor (GLP2R) to modulate energy balance and glucose homeostasis [9], increase in intracellular triacylglycerols (TAG) levels [10], or decrease in ATP and lactate production [26]. Bu contrast, the overexpression of SREBP-1 increased the mRNA levels of lipogenic enzymes, including fatty acid synthase (FAS), elongation of very long-chain fatty acids protein 6 (Elovl6), stearoyl-coenzyme A (CoA), glycerol-3-phosphate acyltransferase 1 [11], fatty acid desaturase 2 (FADS2) [27], and insulin resistance [11]. Thus, given these large number of genes modulated directly or indirectly by SREBP in mammals, it is conceivable that the SREBP homolog in penaeid shrimp could also be involved in the regulation of various important physiological and or pathophysiological functions.

The Pacific white shrimp *P. vannamei* is the most farmed shrimp species globally, accounting for almost 80% of the total produce [22]. Despite the huge production output of *P. vannamei*, these shrimps easily succumb to pathogenic infections, because most of the molecular pathways and mechanisms involved in key physiological and cellular functions are not well understood. The hepatopancreas in decapod crustaceans is the central metabolic organ involved in numerous physiological and pathophysiological functions, and also the major target organ for toxicants and pathogens in the environment [28,29]. Given that in crustaceans the full repertoire of genes modulated by SREBP is unknown, the main aim of this study was to profile SREBP-modulated genes in penaeid shrimp using high throughput sequencing. Thus, the current study used RNA interference-mediated knockdown of SREBP in the hepatopancreas of *P. vannamei* followed by RNA sequencing (RNA-Seq) analysis to profile the genes putatively modulated by SREBP. Gene ontology (GO) and Kyoto Encyclopedia of Genes and Genomes (KEGG) enrichment analyses of the differentially expressed genes (DEGs) identified many candidate genes, a few of which were randomly picked and further validated by quantitative polymerase chain reaction (qPCR) analysis. Besides the genes known to be regulated by SREBPs, i.e., genes involved in lipid metabolism, cellular signaling, and immune response, the majority of the dysregulated genes are unannotated or involved in uncharacterized physiological processes. Thus, these results support the functional diversity of SREBPs, and the large repertoire of genes modulated across species, although most of these genes are currently unannotated in penaeid shrimp.

## 2. Materials and Methods

### 2.1. Experimental Animals and RNA Interference (RNAi) Experiments

Healthy adult *P. vannamei* (10 ± 2 g each) obtained from a local shrimp farm (Shantou Huaxun Aquatic Product Corporation, Shantou, Guangdong, China), were cultured in laboratory tanks containing aerated seawater (1% salinity) at 25 °C and fed once daily with commercial feed (Tongwei Feed Co., Ltd., Xiamen, China). Shrimps were acclimatized to laboratory conditions for 2–3 days before experiments. In the siRNA-mediated knockdown experiments, siRNA targeting the open reading frame (ORF) sequence of *P. vannamei* SREBP (si*Pv*SREBP) and scrambled control siRNA (siNon) were designed, chemically synthesized and high-performance liquid chromatography (HPLC) purified by a commercial company (GenePharma, Suzhou, China). Sixty randomly selected pre-acclimatized shrimp were randomly divided into two groups (30 individuals per group) and the experimental group shrimps were each intramuscularly injected, as previously described [30], with 2.0 µg si*Pv*SREBP, while the control group shrimp were injected with an equivalent amount of siNon. At 48 h post-injection, hepatopancreas samples were collected from three randomly selected shrimp per group and processed for total RNA extraction and cell lysis preparation to ascertain successful knockdown of *Pv*SREBP using quantitative polymerase chain reaction (qPCR) and Western blot analyses. The gene-specific primers used for siRNA synthesis are shown in Table 1. All animal experiments were carried out in accordance with the guidelines and approval of the Animal Research and Ethics Committees of Shantou University, China.

### 2.2. Western Blot Analysis

To ascertain successful knockdown of *Pv*SREBP using Western blot analysis, hepatopancreas cell lysates were prepared and separated by sodium dodecyl sulfate polyacrylamide gel electrophoresis (SDS-PAGE) followed by Western blot analysis. Briefly, shrimp hepatopancreas tissues were homogenized on ice in PBS (0.01 M, pH 7.4) containing 4× phenylmethylsulfonyl fluoride (PMSF) (Beyotime Biotechnology, Shanghai, China), and hepatopancreatic cells collected by centrifugation at 100× *g* (4 °C for 7 min). Cells were washed three times with PBS before being lysed for 20 min at 4 °C with lysis buffer (25 mM HEPES, 150 mM NaCl, 1% Triton X-100, 1 mM EDTA-Na_2_·2H_2_O, PH 7.4) containing a mixture of protease inhibitors (Roche, Indianapolis, IN, USA) and 4x PMSF. Next, cell lysates were centrifuged at 20,000× *g* (4 °C for 20 min) to collect the supernatant before being mixed with 5x loading buffer (42 mmol/L Tris-HCl, containing 100 mL/L glycerol, 23 g/L SDS, 50 g/L 2-mercaptoethanol and 0.02 g/L bromophenol blue) and boiled for 10 min. Samples were then separated on SDS-PAGE and transferred onto polyvinylidene fluoride (PVDF) membranes (Millipore, Billerica, MA, USA) with the Mini Trans-Blot cell wet transfer system (Bio-Rad, Richmond, CA, USA) according to the manufacturer’s instructions. Next, the PVDF membranes were blocked for 2 h at room temperature with 5% skimmed milk dissolved in Tris buffer solution with Tween (TBST) (20 mM Tris, 150 mM NaCl, 0.1% Tween 20, pH 7.6), followed by incubation at room temperature with 1:1000 dilution of mouse anti-*Pv*SREBP (produced in-house) or 1:1000 dilution of mouse anti-tubulin (Sigma-Aldrich, St Louis, MO, USA) primary antibodies for 2 h. After being washed three times (15 min each) with TBST, membranes were incubated at room temperature with 1:3000 dilution of horseradish peroxidase (HRP)-linked goat anti-mouse secondary antibodies (Sigma-Aldrich, St Louis, MO, USA) for 2 h. Finally, signals were detected by chemiluminescence using enhanced chemiluminescence (ECL) reagent (Millipore, Billerica, MA, USA), and developed by the Amersham Imager 600 (GE, Boston, MA, USA).

### 2.3. RNA Extraction, cDNA Synthesis, and qPCR Analysis

Total RNA was extracted from pooled hepatopancreas samples of three shrimps per treatment using Trizol reagent (Invitrogen, Carlsbad, CA, USA) according to the manufacturer’s instruction. The RNA concentration and purity were measured using a NanoDrop 2000 spectrophotometer (Nano-drop Technologies, Waltham, MA, USA), while RNA quality was ascertained by the A260/280 ratio (1.8–2.2) and 1% agarose gel electrophoresis. All RNA samples were used immediately or stored at −80 °C in aliquots for later use. Only high-quality RNA samples from three biological replicates were used to construct the cDNA libraries. For cDNA synthesis, 1.0 µg total RNA was used with the TransScript™ One-step gDNA removal and cDNA Synthesis SuperMix kit (TransGen Biotech, Beijing, China), following the manufacturer’s protocol. The cDNA samples were used immediately or stored at −20 °C in aliquots for later use. In the qPCR analysis, the reaction mixture contained 2× RealStar Green Power Mixture (GenStar, Beijing, China), 1 µL cDNA template, 0.5 µL each of forward and reverse primers (10 µM), plus ddH_2_O to a total volume of 20 µL. Triplicate samples per treatment were analyzed on a qTOWER 3G Real-Time PCR system (Analytik Jena AG, Überlingen, Germany) with the following program: one cycle at 95 °C for 10 min and 45 cycles of 95 °C for 15 s and 60 °C for 30 s. The EF1α gene of *P. vannamei* (*Pv*EF1α) was used as the housekeeping gene, and the relative gene expression computed using the 2^−ΔΔCT^ method [31] normalized to *Pv*EF1α. The gene-specific primers used are shown in Table 1.

### 2.4. Library Construction and Transcriptome Sequencing

The two cDNA libraries (experimental and control) used for the transcriptome sequencing were prepared with 1 µg total RNA (pooled from three independent biological samples per treatment) using the Truseq^TM^ RNA sample prep Kit (Illumina, San Diego, CA, USA) before being sequenced by a commercial company (Shanghai Majorbio Bio-pharm Technology Co., Ltd. Shanghai, China) on the Illumina Novaseq 6000 platform. Briefly, poly-A containing mRNA was isolated from total RNA using oligo (dT) beads, before being fragmented and reverse transcribed into double-stranded cDNA using random primers (Illumina, San Diego, CA, USA). Next, sequencing adapters were attached to the short cDNA fragments and PCR amplified (15 PCR cycles) followed by selection on 2% agarose gel. After quantification by TBS380 (Picogreen, Invitrogen, Carlsbad, CA, USA), the paired-end libraries were sequenced on the Illumina Novaseq 6000 platform [32]. The assembled sequence data from this article has been submitted to GenBank under accession number PRJNA756609.

### 2.5. Transcriptome Data Analysis, Functional Annotation, and qPCR Validation

The raw paired-end reads generated by the Illumina Novaseq 6000 platform were filtered and trimmed using the SeqPrep (https://github.com/jstjohn/SeqPrep accessed on 13 July 2021) and Sickle (https://github.com/najoshi/sickle accessed on 13 July 2021) programs. Clean reads were aligned to the *Penaeus_vannamei* reference genome (ASM378908v1) at NCBI (https://www.ncbi.nlm.nih.gov/genome/10710?genome_assembly_id=422001 accessed on 12 October 2022) using TopHat2 v2.1.1 (http://tophat.cbcb.umd.edu/ accessed on 31 August 2021) and/or HISAT2 v2.1.0 (http://ccb.jhu.edu/software/hisat2/index.shtml accessed on 31 August 2021). Unigenes were identified by BLASTx search and annotated to six protein databases, including the non-redundant (NR) and Cluster of Orthologous Group (COG) protein database at GenBank NR v2019.6.26 (ftp://ftp.ncbi.nlm.nih.gov/blast/db/ accessed on 31 August 2021), Swiss-Prot v2019.7.1 (http://www.expasy.ch/sprot accessed on 31 August 2021), Kyoto Encyclopedia of Genes and Genomes (KEGG) v2020.03 (http://www.genome.jp/kegg accessed on 31 August 2021), Gene Ontology (GO) GO v2019.7.1 (http://www.geneontology.org/ accessed on 31 August 2021), Pfam version Rfam v14.1 (http://rfam.janelia.org/ accessed on 31 August 2021), and RSEM v1.3.1 (http://deweylab.biostat.wisc.edu/rsem/ accessed on 31 August 2021). Differentially expressed genes (DEGs) were calculated using transcripts per million reads (TPM), while differential expression analysis used DEseq2 v1.24.0 (http://bioconductor.org/packages/stats/bioc/DESeq2/ accessed on 31 August 2021). Unigenes with Benjamini Hochberg (BH) |log2FC| ≥ 1 and *p* < 0.05 were chosen as the DEGs. Gene Ontology (GO) enrichment analysis was performed on DEGs using Goatools v0.6.5 (https://github.com/tanghaibao/GOatools accessed on 31 August 2021) and also for multiple tests of BH to correct for the *p*-values. The GO function was considered significantly enriched when the corrected *p*-value (False Discovery Rate, FDR) was <0.05. KEGG pathway enrichment analysis was performed on DEGs using an R script and KEGG pathway function was considered significantly enriched when the corrected *p*-value (Pvalue_uncorrected) was <0.05. Biorender online website (https://biorender.com/ accessed on 29 September 2021) and GraphPad Prism software (v8.0.2) were used to plot the charts and graphs. Heatmaps showing the expression levels for selected genes were constructed with TBtools software v1.098696 (https://github.com/CJ-Chen/TBtools accessed on 29 September 2021) after log2 transformation.

To validate the RNA-seq results using qPCR, the following nine DEGs were selected: FABP, NOS1, MNK, NFκBIA, C-type lectin, PFKFB2, CREB3, COX2, and HK for validation. Gene-specific primers (see Table 1) were designed using Primer 5.0 software and the qPCR was performed in triplicates as described above in Section 2.3.

### 2.6. Protein-Protein Interaction Network Construction

A protein-protein interaction (PPI) network of the identified DEGs (353 genes) was constructed through the STRING database v11.5 (https://cn.string-db.org/ accessed on 28 November 2021) and the Cytoscape software v3.7.1 (https://cytoscape.org/ accessed on 28 November 2021). The minimum required interaction score was ≥0.4. The cytoHubba plugin in Cytoscape was used to select the top 10 ranked nodes in the PPI network, while the molecular complex detection (MCODE) algorithm was used to screen subnetworks (degree cutoff = 2, node score cutoff = 0.2, k-core = 2, and maximum depth = 100), as previously described [33].

### 2.7. Shrimp Survival

To determine shrimp survival rate after *Pv*SREBP knockdown, shrimp (30 individuals per group) were injected with si*Pv*SREBP or siNon as described above in Section 2.1. At 6 h intervals, the number of dead shrimps in each group was counted and recorded. Shrimp survival rate was analyzed using the Kaplan-Meier estimate [34], and the significance was compared using the log-rank test [35] in GraphPad Prism 8.

### 2.8. Statistical Analysis and Data Presentation

Data were analyzed and subjected to a one-way analysis of variance (one-way ANOVA) followed by an unpaired two-tailed *t*-test with significance considered at *p* < 0.05 and presented as mean ± standard deviation (SD). All statistical analyses used Microsoft Excel 2016 Student Edition. Charts and graphs were drawn using 8.0.2 for Windows, GraphPad Software, San Diego, CA, USA (www.graphpad.com accessed on 12 Octomber 2022), bioinformatics online website (http://www.bioinformatics.com.cn/ accessed on 28 November 2021), Venny 2.1 online website (https://bioinfogp.cnb.csic.es/tools/venny/index.html accessed on 28 November 2021), and TBtools software v1.098696 (https://github.com/CJ-Chen/TBtools/releases accessed on 28 November 2021).

## 3. Results

### 3.1. Transcriptome Profiling, and Analysis of Differentially Expressed Genes

To prepare the two RNA libraries for sequencing on the Illumina Novaseq 6000 platform, we first ascertained the successful knockdown of *Pv*SREBP in shrimp hepatopancreas using Western blot and quantitative polymerase chain reaction (qPCR) (Figure 1A). RNA sequencing generated a total of 337,337,742 raw reads and after the removal of adaptor primers and low-quality and very short reads, 334,855,446 clean reads were obtained (Table 2). After assembly, 41,345 transcripts were obtained for the two groups, with 23,442 unigenes identified. For the functional annotation of the identified genes, 16,084 unigenes were aligned to the Pfam, NR, Swiss-Prot, KEGG, COG, and GO protein databases using BlastP (Table 3).

Analysis of the significantly (*p* < 0.05) expressed genes in the two libraries revealed that 11,972 genes (95.0%) were expressed in the *Pv*SREBP knockdown group, including 493 genes exclusively expressed in the si*Pv*SREBP group. On the other hand, 12,112 genes (96.1%) were expressed in the control (siNon) group, with 633 genes only expressed in the siNon group. The number of genes expressed in both the si*Pv*SREBP and siNon groups was 11,479 genes (Figure 1B) and distributed as shown in Figure 1C,D. When the differentially expressed genes (DEGs) from the significantly expressed unique and shared genes were screened using a threshold of |log2 fold change| ≥ 1 and *p* < 0.05, 706 DEGs were identified, among which 282 DEGs (39.9%) were upregulated, and 424 (60.1%) were downregulated (Figure 1E).

### 3.2. Functional Annotation of DEGs

When the protein orthologs of DEGs were analyzed, 552 DEGs were successfully annotated into 20 clusters of orthologous groups (COGs). Besides function unknown (267 genes), the largest category was posttranslational modification, protein turnover, and chaperones (56 genes), followed by intracellular trafficking, secretion, and vesicular transport (36 genes), and carbohydrate transport and metabolism (35 genes). The functional categories with the least annotations were RNA processing and modification (three genes), chromatin structure and dynamics (three genes), and cell cycle control, cell division, and chromosome partitioning (three genes) (Figure 2A).

Gene ontology (GO) functional classification was used to successfully assign 290 DEGs into 25 GO terms (Figure 2B), distributed into three categories, i.e., biological process (106 genes), cellular component (92 genes), and molecular function (92 genes). In the biological process category, most of the DEGs were distributed in the metabolic process (46 genes) and cellular process (40 genes) subcategories, while in the cellular component category, most DEGs were in the cell part (36 genes), membrane (16 genes), organelle (10 genes), organelle part (nine genes), and membrane part (nine genes) subcategories. On the other hand, most of the DEGs in the molecular function category were distributed in the catalytic activity (55 genes) and binding (26 genes) subcategories. When functional enrichment analysis was used to further annotate the DEGs into the top 20 enriched GO terms, the highly enriched GO categories were catalytic activity (56 genes), organonitrogen compound metabolism (29 genes), small molecule metabolism (23 genes), organic substance biosynthesis (18 genes), carboxylic acid metabolism (16 genes), oxoacid metabolism (16 genes), and organic acid metabolism (16 genes) (Figure 2C).

### 3.3. Pathway Functional Analysis of DEGs

When KEGG pathway classification and functional enrichment analyses were used to explore the biological pathways of the DEGs, 654 DEGs were mapped to six major KEGG pathways, i.e., organismal systems, human diseases, metabolism, environmental information processing, cellular processes, and genetic information processing (Figure 3A). Most of the DEGs enriched in the KEGG pathway are downregulated, especially genes involved in signal transduction, endocrine system, cancer, infectious diseases, digestive system, and the immune system. Among the top 20 KEGG-enriched pathways (Figure 3B), 147 DEGs were highly enriched in metabolism (76 genes), organismal systems (55 genes), and human diseases (16 genes) (Appendix A). The highest enriched KEGG pathways include pancreatic secretion, drug metabolism-other enzymes, human cytomegalovirus infection, amino sugar and nucleotide sugar metabolism, cholesterol metabolism, cholinergic synapse, and Toll and lmd signaling pathways (Figure 3B).

### 3.4. DEGs with Annotated and Unannotated Functions in Penaeid Shrimp

To identify the physiological and or pathophysiological functions of the DEGs dysregulated after *Pv*SREBP knockdown, genes with *p* < 0.05 and | log2 (fold change) | ≥ 2 were screened (Appendix A). We found that many of the significantly downregulated DEGs were annotated in immune response (Figure 4A), energy metabolism (Figure 4B), and signal transduction pathways (Figure 4C). Interestingly, the majority of the significantly downregulated DEGs are unannotated (Figure 4D and Appendix A). Nonetheless, many of these unannotated genes, such as LOC113818648, LOC113820682, LOC113800798, LOC113807644, LOC113811284, were predicted using six databases (i.e., GO, KEGG, COG, NR, Swiss-Prot, and Pfam) and found to be putatively involved in several key cellular functions (Appendix A). To ascertain the RNA-seq data, nine randomly selected DEGs i.e., MNK, NFκBIA, C-type lectin, FABP, PFKFB2, CREB3, COX2, HK, and NOS1 were validated using qPCR analysis, and their expression followed similar patterns as the RNA-seq results (Figure 5).

### 3.5. Protein-Protein Interaction Network between DEGs

To determine the relationship between the DEGs, we examined the protein-protein interaction (PPI) network using the STRING database. The results (Appendix A) revealed 180 nodes and 31 edges, with an average node degree of 0.344 and an average local clustering coefficient of 0.0986, indicating that the PPI enrichment was statistically significant (Figure 6A). Ten top hub genes were identified by CytoHubba among 29 DEGs, i.e., glycogen phosphorylase-like, phosphoserine phosphatase-like, D-3-phosphoglycerate dehydrogenase-like, argininosuccinate synthase-like, NF-kappa-B inhibitor cactus-like, mannose-1-phosphate guanyltransferase β-like, β-glucuronidase-like, 4F2 cell-surface antigen heavy chain-like, T-complex protein 1 subunit γ-like, and large neutral amino acids transporter small subunit 2-like (Figure 6B). Among these hub genes, a significant subnetwork, comprising glycogen phosphorylase, phosphoserine phosphatase, D-3-phosphoglycerate dehydrogenase, and argininosuccinate synthase was found to interact independently within the PPI network with an MCODE score of four and six nodes (Figure 6C).

### 3.6. Effect of PvSREBP Knockdown on Shrimp Survival

Given that a correlation exists between SREBP-1 expression and survival in human hepatocellular carcinoma patients [36], whereas SREBP-2 is reported to be critical for survival and limb patterning during mice development [37], we explored whether *Pv*SREBP has any effects on shrimp survival. When *Pv*SREBP was knocked down and shrimp survival rates were determined compared with control (siNon), no significant difference was observed between the survival rates of si*Pv*SREBP and siNon groups (Figure 7). However, after 36 h of knockdown, the si*Pv*SREBP group shrimp had lower survival rates compared with control (siNon), although the difference was not significant. These results suggest that the siRNA targeting *Pv*SREBP does not have an off-target effect or did not affect genes involved in survival or other physiological functions that are directly linked with shrimp survival.

## 4. Discussion

As versatile transcription factors, sterol regulatory-element binding proteins (SREBPs) can integrate multiple cellular signals to control lipogenesis and important pathways involved in diverse biological processes, including endoplasmic reticulum (ER) stress, inflammation, autophagy, and apoptosis [38]. Recent studies have also shown that the SREBPs play essential regulatory roles in immune response and lipid metabolism in vertebrates, mollusks, and crustaceans [19,21,23]. The liver plays a central role in lipid metabolism in mammals, hence studies on SREBPs are mostly performed using the liver and adipose tissues [39,40]. In decapod crustaceans the hepatopancreas is the central metabolic organ involved in numerous functions, such as absorption and metabolism of nutrients, synthesis of digestive enzymes, hemolymph proteins, and immune effectors, detoxification of xenobiotics, storage of energy reverses, and major target organ for toxicants and pathogens in the environment [28,29]. The hepatopancreas performs synonymous functions with the mammalian liver, and is a suitable organ to use in exploring the diverse regulatory functions of SREBP in *P. vannamei*. Although the SREBP of some decapod crustaceans have been characterized, most of the genes or physiological processes modulated by SREBP are unknown. Thus, in the current study, we used RNAi followed by transcriptome analysis to reveal that besides the genes involved in immune response and lipid metabolism, the SREBP homolog in *P. vannamei* (*Pv*SREBP) also modulates many genes involved in energy metabolism and signal transduction pathways, with the majority of the dysregulated genes unannotated.

In this study, 706 DEGs were identified, with most of them (i.e., 424) DEGs (60.1%) downregulated. When the DEGs were subjected to gene ontology (GO) analysis, most were enriched in terms associated with catalytic activity and metabolism, while KEGG pathway enrichment analysis showed that the enriched DEGs are mainly involved in key physiological processes, including signal transduction, endocrine system, digestive system, immune system, environmental adaptation, and infectious diseases. (Figure 3). Penaeid shrimp, like other marine invertebrates, leverage a large repertoire of factors to augment their innate immune response (recently reviewed by [41]. Consistent with our previous study [23], many genes involved in immune response, such as trypsin, and C-type lectin, and fatty acids metabolism-related genes, including cyclooxygenases (COX), fatty acid synthase (FASN), and FABP were dysregulated. But most importantly, genes with multiple functions or involved in cellular signaling pathways, such as the mitogen-activated protein kinase (MAPK)-interacting kinase (MNK) and vascular endothelial growth factor (VEGF), were dysregulated after *Pv*SREBP knockdown, while the majority of the significantly dysregulated genes are unannotated.

In crustaceans the hepatopancreas plays a central role in metabolism, hence it is not surprising that genes involved in metabolism, especially energy metabolism were dysregulated after *Pv*SREBP knockdown. For instance, in glucose metabolism, the key enzyme, hexokinase (HK), catalyzes the first rate-limiting step that converts glucose into glucose-6-phosphate [42]. In this study, HK was downregulated (Figure 5). Although the role of HK has not been demonstrated in penaeid shrimp, in different mammalian cell models, insulin induces the expression of SREBP-1c to modulate the expression of glycolytic and lipogenic enzyme genes [43], including HKII [44], one of the four hexokinase isoforms in mammalian tissues [42]. Indeed, the effect of insulin on the transcriptional effect of SREBP-1c on the human HKII gene, which contains a sterol regulatory element (SRE) domain on its promoter, has been demonstrated in muscle cells [45]. Moreover, the knockdown of SREBP1 and SREBP2 reduced both the glycolytic potential and mitochondria-mediated oxidative phosphorylation in colon cancer cells, suggesting that SREBP-dependent lipid biogenesis has an important role in maintaining overall metabolic activity in cells [46]. In glucose metabolism, the bifunctional enzyme 6-phosphofructo-2-kinase/fructose-2,6-bisphosphatase (PFKFB) is another important gene that was found to be dysregulated in this study. PFKFB catalyzes the synthesis and degradation of fructose-2,6-bisphosphate, a key modulator of glycolysis-gluconeogenesis [47]. Thus, the observed downregulation of both HK2 (hexokinase type 2) and PFKFB (Figure 5) is consistent with previous studies where SREBP enhanced glycolysis by activating PFKFB [48]. After all, in the metabolism of glucose to cytosolic citrate in cytokine-stimulated NK cells, SREBP is required for glycolysis and oxidative phosphorylation, suggesting that SREBP-dependent metabolism is critical for cellular immune responses [4] not only in mammals but also in crustaceans. In this study, the involvement of SREBP in glucose metabolism in shrimp could further be inferred from the PPI network, which revealed a strong association with glycolysis (Figure 6). It is, therefore, conceivable that SREBP knockdown attenuated the expression of genes involved in metabolism because silencing of SREBP1 or SREBP2 alters cellular metabolism by reducing mitochondrial respiration, glycolysis, and fatty acid oxidation [46].

In mammals, the SREBP pathway regulates diverse cellular processes, including phagocytosis, cell cycle progression, oxygen sensing, and survival in response to bacterial infection [49]; however, such regulation has not been reported in penaeid shrimps or crustaceans. Thus, it was intriguing to observe that several of the dysregulated genes, such as MNK and VEGF, are involved in cellular signal transduction pathways that modulate many key metabolic and physiological processes. MNKs are serine/threonine protein kinases that are activated by the ERK1/2 (extracellular regulated kinase) and p38α/βMAPK pathways [50]. Humans and mice express two MNK proteins, MNK1 and MNK2, encoded by separate genes, *Mknk1* and *Mknk2*, respectively. Mammalian MNK1/2 can phosphorylate eIF4E (eukaryotic initiation factor 4E) [51,52] to modulate cell proliferation and inflammation [53] or phosphorylate cytosolic phospholipase A_2_ (cPLA2) [54] to release arachidonic acid for the production of eicosanoids [55]. Thus, besides their catalytic activity, MNKs regulate metabolism in adipocytes by modulating the expression of genes involved in de novo lipogenesis and triglyceride syntheses, such as carbohydrate response-element binding protein (ChREBP), SREBP, and hormone-sensitive lipase [50]. The observed downregulation of MNK in this study is, therefore, quite intriguing and requires further examination. Another noteworthy gene that was downregulated in this study was nuclear factor of kappa light polypeptide gene enhancer in B-cells inhibitor α (NFκBIA). Although members of the NF-kappaB family, including cactus, a homolog of IκB (Ikappa B α), have been identified and characterized in *Drosophila* [56] and *P. vannamei* [57], their role in crustaceans metabolism is unknown. Thus, attenuation of NFκBIA expression in this study upon *Pv*SREBP knockdown is an interesting observation, which could be further explored to delineate the relationship between SREBP and the NF-kappaB transcription factor, since NF-kappaB regulates diverse genes involved in immune function, growth, cell proliferation, apoptosis, and metastases, while its activity is inhibited by NFκBIA [58].

In mammals, the membrane-bound transcription factor, CREB3 (cAMP responsive element-binding protein 3), is involved in the regulation of liver and small intestine lipid metabolism and controls hepatic triglyceride [59,60] and glucose metabolism through the activation of plasma fibroblast growth factor 21 (FGF21) and lipoprotein lipase [61]. Among the five CREB3 family members (i.e., CREB3, CREB3L1, CREB3L2, CREB3L3, and CREB3L4) described in mammals [62], CREB3 and SREBP share a similar cleavage system for nuclear transactivation in the endoplasmic reticulum (ER), and therefore functionally inhibit each other [63,64] to regulate hepatic lipogenesis [65]. Here, the observed downregulation of CREB3 expression suggest that the relationship between SREBP and CREB3 in penaeid shrimp or crustaceans might be different, and requires further studies to delineate. Fatty acid-binding protein (FABP) is another key factor involved in energy metabolism that was dysregulated in this study. As ubiquitously expressed intracellular lipid chaperones that coordinate lipid trafficking and signaling in cells, some FABP isoforms are linked to metabolic and inflammatory pathways in mammals [66]. Thus, the increased expression of FABP observed in this study suggests that its role in penaeid shrimp could be different from mammals or under different pathophysiological conditions, given that the expression of FABP3 and FABP7 in cancer cells decreased under various conditions, such as hypoxia, when either SREBP1 or SREBP2 was depleted [67].

Our current data indicate that *Pv*SREBP modulates many more genes in penaeid shrimp than previously thought. For instance, the majority (57.80%) of the downregulated DEGs are unannotated, although gene orthology prediction reveals that most of these unannotated genes play putative roles in various physiological and pathophysiological functions. For instance, the gene LOC113808628 is predicted as CTLDcp2 (in NR databases) or Lectin C-type domain/Low-density lipoprotein receptor domain class A (in Pfam databases), with gene LOC113802130 predicted as caspase 3, whereas genes LOC113800809 and LOC113800798 are predicted as low-density lipoprotein receptor-related protein 2 (Figure 4 and Appendix A). Similarly, gene LOC113820682 has been predicted as a heat shock protein (HSP), while LOC113818648 and LOC113800388 are predicted as G-protein coupled receptors (GPCRs). Given that the inhibition of HSP90 can destabilize the precursor forms of SREBP [68] to attenuate the expression of its downstream genes, such as FAS [69], whereas GPCRs are candidate receptors for members of the crustacean hyperglycemic hormone (CHH) superfamily [70], we contend that *Pv*SREBP might play regulatory roles in ecdysteroidogenesis, growth, molting, and many other physiological functions through CHH modulation. Indeed, the gene LOC113811284 was up-regulated after *Pv*SREBP knockdown, and since this gene is predicted as β-N-acetylglucosaminidase, a key enzyme involved in chitin degradation [71], it further supports the importance of *Pv*SREBP in molting. From the foregoing, it is clear that SREBP modulates as many genes in shrimp as in mammals, but the functions of most of the modulated genes in shrimp are unknown partly because there has been limited research on SREBP in shrimp or because the complete genome of *P. vannamei* has just been sequenced [22]; therefore, most of the genes have not been annotated. Thus, the true repertoire of SREBP-modulated genes in penaeid shrimp would only become clear when the annotation of the *P. vannamei* genome data is complete or when more studies on SREBP have been carried out. Collectively, the findings from this study reveal that SREBP modulates many genes involved in crucial physiological processes, including energy metabolism, immune responses, and cellular signaling pathways, while the majority of the dysregulated genes are unannotated in penaeid shrimp.

## 5. Conclusions

Our current data shows that the SREBP homolog in penaeid shrimp modulates the expression of many genes, including genes involved in immune responses, energy metabolism, and signal transduction pathways. Most importantly, most of the genes modulated by SREBP have multiple cellular functions, while the majority of the downregulated genes are unannotated, indicating that the full repertoire of *Pv*SREBP-regulated genes and their associated physiological functions in penaeid shrimp remains enigmatic.

## Figures and Tables

**Figure 1 genes-13-02057-f001:**
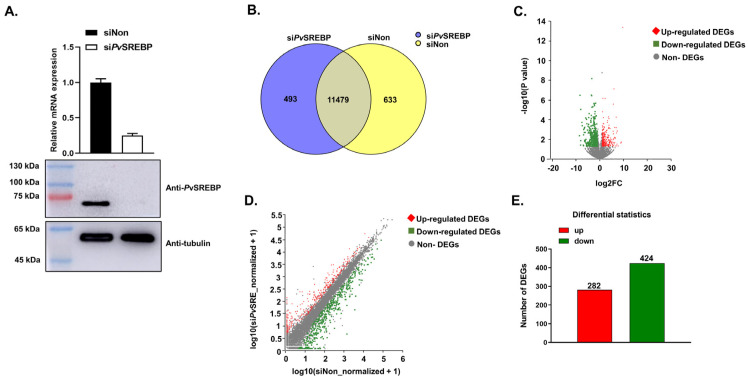
*Pv*SREBP knockdown and the distribution of differentially expressed genes (DEGs). (**A**) Knockdown efficiency of *Pv*SREBP determined by qPCR and Western blot analyses. Data represent mean ± SEM (*n* = 3). Statistical significance considered at *p* < 0.05. The immunoblots shown are representative of three independent experiments. (**B**) Number of significantly expressed genes in the two transcriptome libraries. (**C**) Distribution of upregulated and downregulated DEGs. (**D**) Relationship between upregulated and downregulated DEGs. (**E**) Number of upregulated and downregulated DEGs.

**Figure 2 genes-13-02057-f002:**
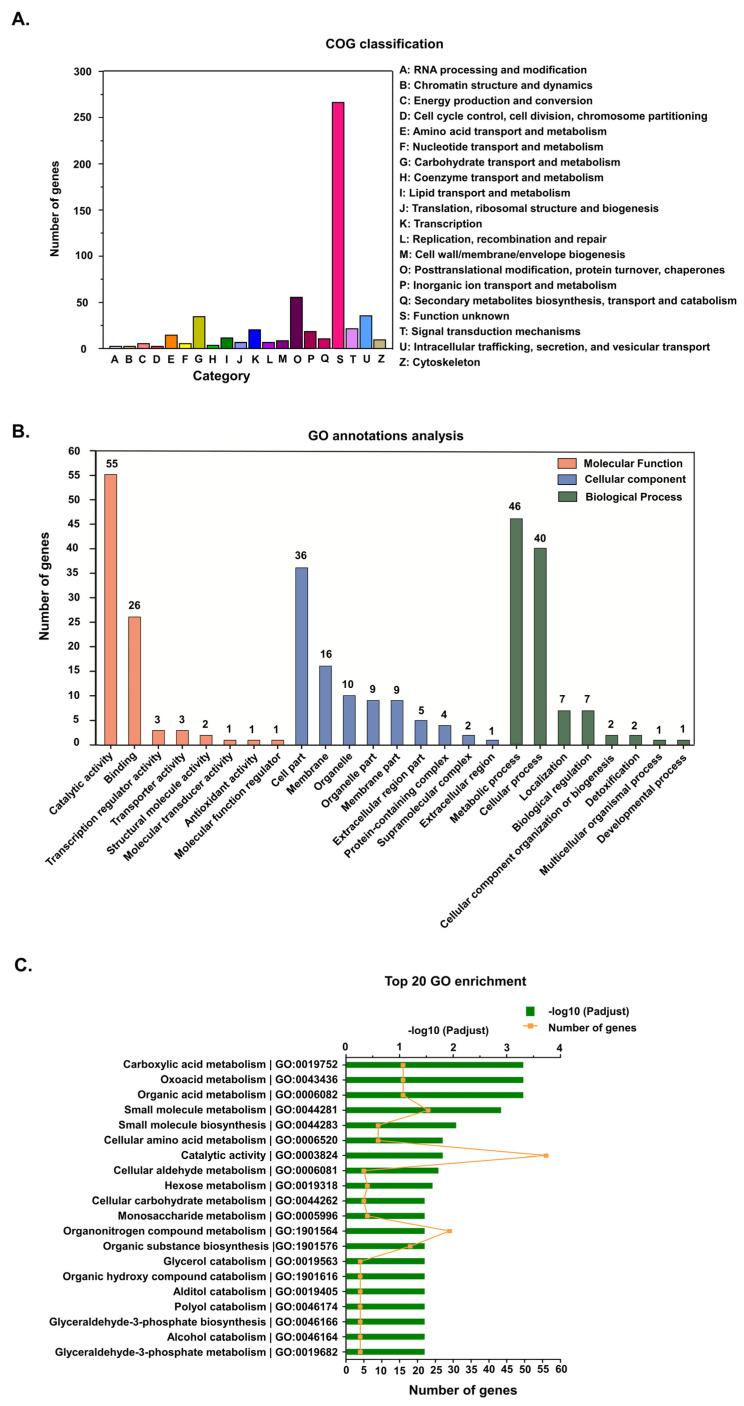
Annotation of differentially expressed genes (DEGs). (**A**) Cluster of orthologous groups (COG) classification of the putative proteins encoded by the DEGs. (**B**) Gene ontology (GO) annotations of DEGs. Genes were categorized into three categories: biological process, cellular component and molecular function. (**C**) The top 20 GO enriched DEGs.

**Figure 3 genes-13-02057-f003:**
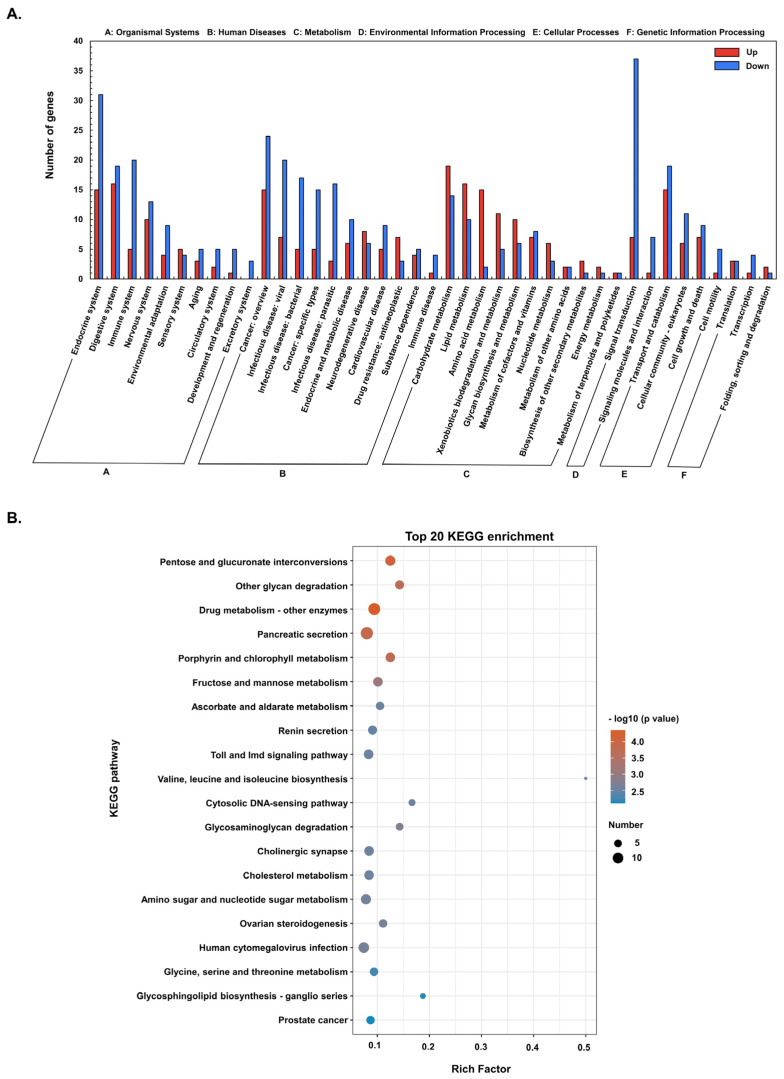
The Kyoto Encyclopedia of Genes and Genomes (KEGG) annotations and enrichment analysis of DEGs. (**A**) KEGG classification of upregulated and downregulated DEGs. (**B**) Top 20 KEGG enriched DEGs.

**Figure 4 genes-13-02057-f004:**
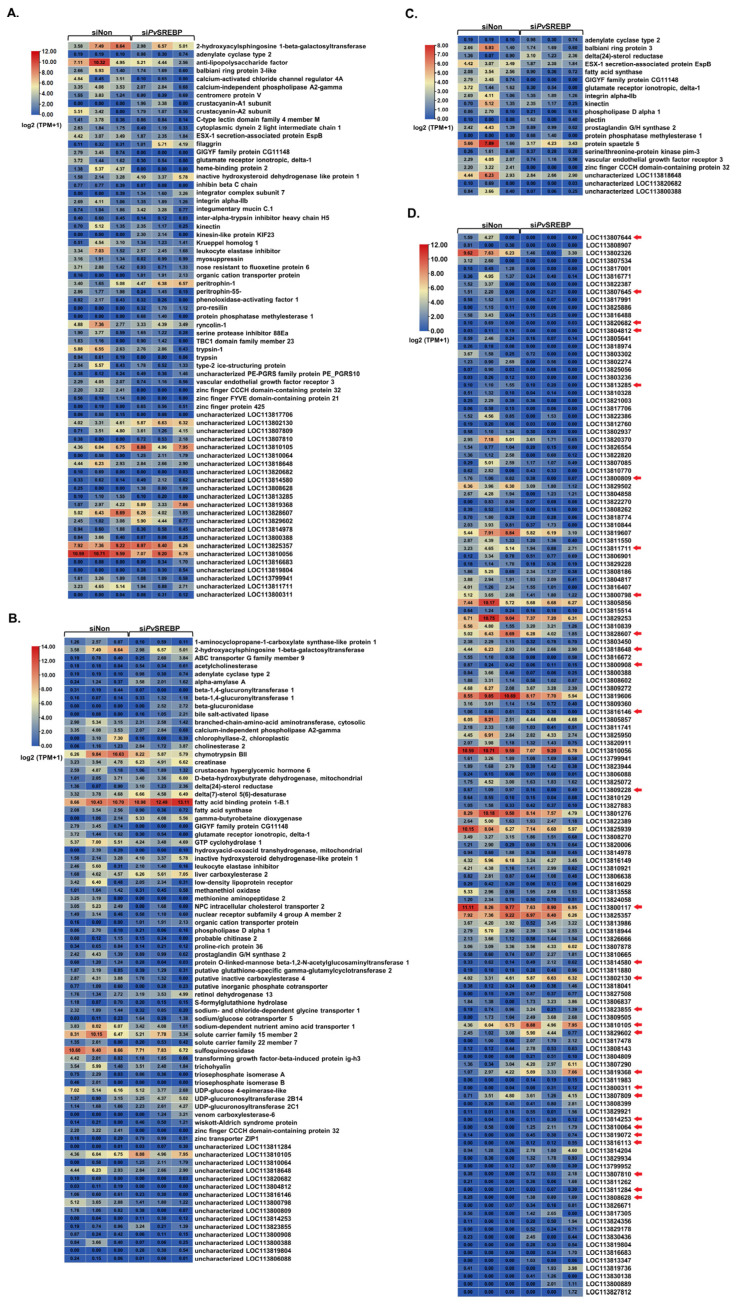
Expression and distribution of important DEGs. Heat maps showing DEGs related to (**A**) immune response, (**B**) metabolism, (**C**) cellular signaling pathways, and (**D**) DEGs with unannotated functions. The red arrows point to genes with putatively annotated functions in the NR database. Three independent samples were analyzed per treatment (siNon and si*Pv*SREBP). Color legend is on a log2 (TPM + 1) scale.

**Figure 5 genes-13-02057-f005:**
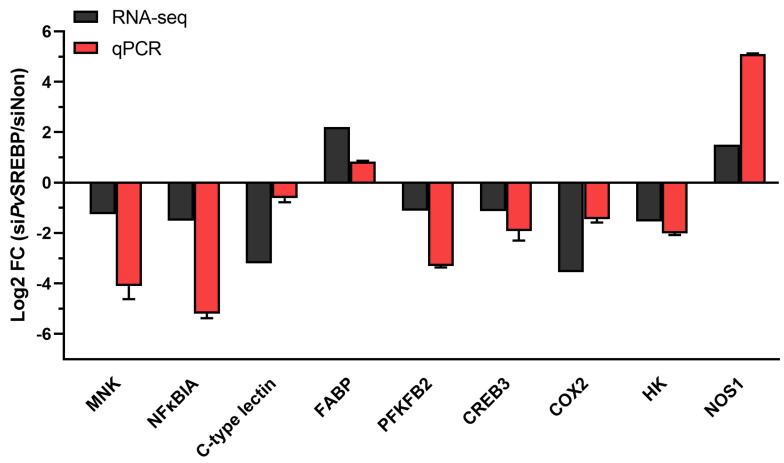
Validation of selected DEGs using qPCRs. The relative expression (fold-change) of nine DEGs was determined by qPCR relative to the expression of the *Pv*EF1α gene (internal control). MNK, MAP kinase-interacting serine/threonine-protein kinase 1; NFκBIA, Nuclear factor of kappa light polypeptide gene enhancer in B-cells inhibitor, α; C-type lectin; FABP, fatty acid binding protein; PFKFB2, 6-phosphofructo-2-kinase/fructose-2,6-bisphosphatase; CREB3, cyclic AMP response element-binding protein A; COX2, Cyclooxygenase-2 (prostaglandin G/H synthase 2); HK2, hexokinase type 2; NOS1, nitric oxide synthase 1.

**Figure 6 genes-13-02057-f006:**
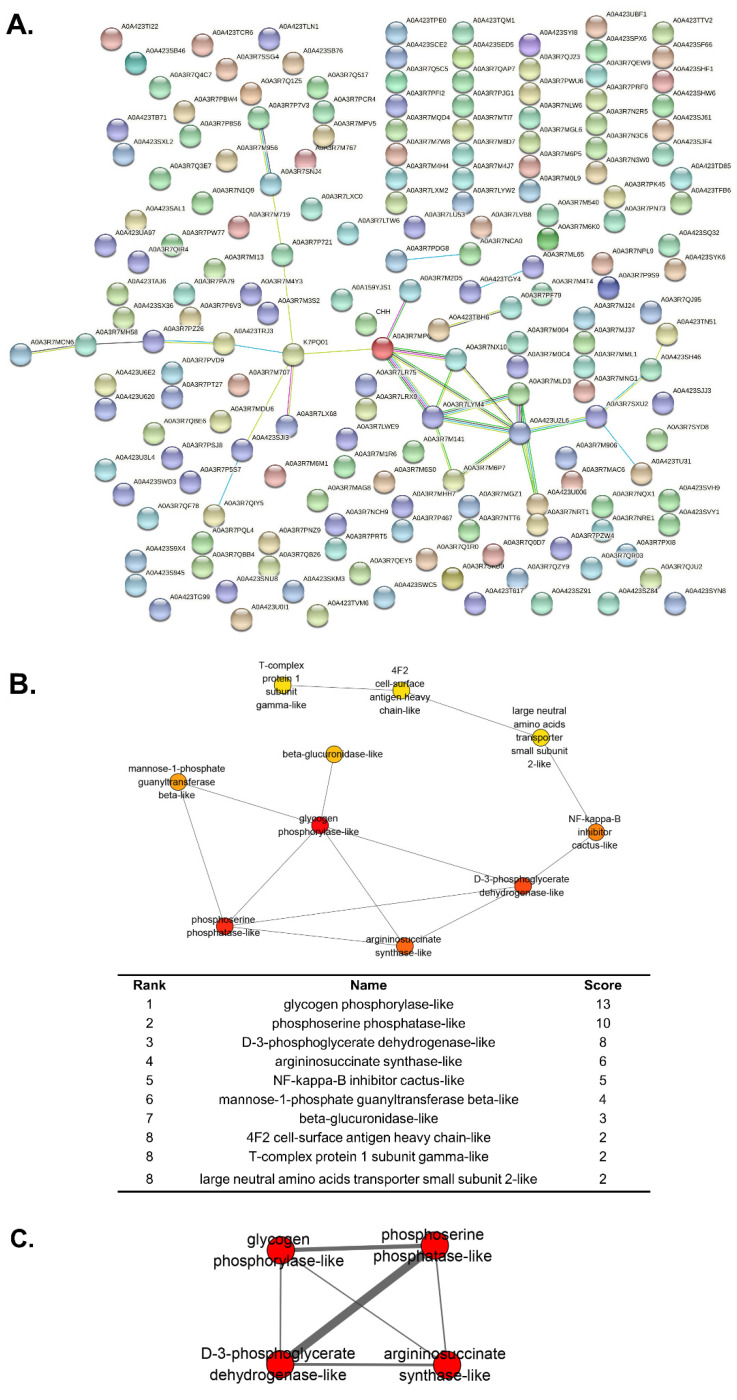
Protein–protein interaction (PPI) network. (**A**) STRING analysis of the identified DEGs. (**B**) The top 10 Hubba nodes ranked by maximum clique centrality (MCC). (**C**) Subnetworks with MCODE score of 4.

**Figure 7 genes-13-02057-f007:**
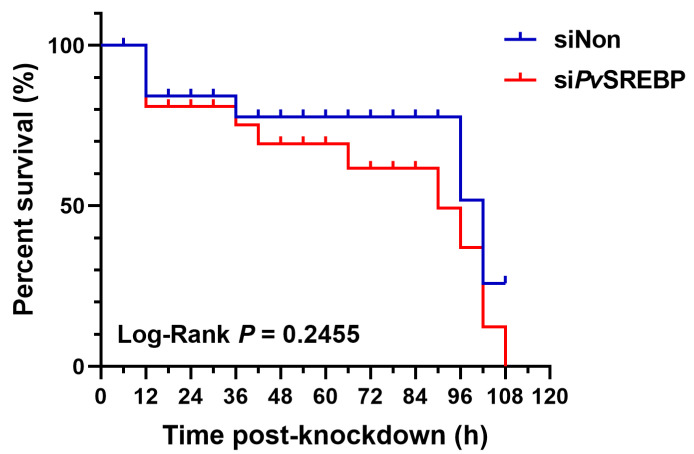
Shrimp (*P. vannamei*) survival curve after *Pv*SREBP knockdown. Shrimp survival (*n* = 30 per group) was determined after intramuscular injection with siNon (scrambled siRNA) si*P*vSREBP and the number of shrimps recorded at 6 h intervals. The product-limit method of Kaplan-Meier was used to calculate the shrimp survival rate and the significance compared using the log-rank test.

**Table 1 genes-13-02057-t001:** The gene-specific primer list.

Primer Name	Sequence (5′-3′)	Amplicon (bp)
**PCR**		
*Pv*SREBP-F	CCATGGCTGATATCGGATCCATGAACTGGCCTGACCTGGACT	1296
*Pv*SREBP-R	TGGTGGTGGTGGTGCTCGAGTTAGTCAGCCATGGAACGTGCC	1296
**Real-time RT-PCR**		
q-*Pv*SREBP-R	GGAGTTGTTGTTGCCGTGG	134
q-*Pv*SREBP-F	TGGCTGAGATGTTGGTAATGG	134
q-MNK-R	ATGCACGACTCGGCGAACAGC	109
q-MNK-F	ACCATCCCTGGGTCAAGAACG	109
q-NFκBIA-R	GTGCCGTCCGACCACTCTT	140
q-NFκBIA-F	TGCCGCTGACCTTACCAAC	140
q-FABP-R	CTCCTCGCCGAGCTTGATGGT	103
q-FABP-F	CGCTAAGCCCGTGCTGGAAGT	103
q-PFKFB2-R	CAAAGACAGCCACTTCACCC	200
q-PFKFB2-F	CCTCAACTGGATCGGCATAA	200
q-CREB3-R	TGGACAGGAAAGCCGTAGCA	218
q-CREB3-F	GAACAACACCGCACCCACCC	218
q-lectin-R	TGATTCCTCGCTCGCCCTAC	120
q-lectin-F	CGCTCTTGCTGTCTGCCTGAT	120
q-COX2-R	GTAGGCATTGAGGGTGATGTAG	103
q-COX2-F	CCACAAGCGACTGATGACTTA	103
q-HK-R	AGCCCATCACCAGGTCCAAT	205
q-HK-F	AGTCCAACCCAGAGGCAACC	205
q-NOS1-R	TCTCTCCCAGTTTCTTGGCGT	104
q-NOS1-F	GAGCAAGTTATTCGGCAAGGC	104
*Pv*EF-1α-R	CCTTTTCTGCGGCCTTGGTAG	118
*Pv*EF-1α-F	TATGCTCCTTTTGGACGTTTTGC	118
**siRNA**		
si*Pv*SREBP-R	UUACGGUGUCGCCAGAAGCTT	21
si*Pv*SREBP-F	GCUUCUGGCGACACCGUAATT	21
siNon-R	ACGUGACACGUUCGGAGAATT	21
siNon-F	UUCUCCGAACGUGUCACGUTT	21

**Table 2 genes-13-02057-t002:** Summary of RNA-seq data for *P. vannamei* injected with siNon or siPvSREBP.

Samples	siNon			si*Pv*SREBP			All
	Sample 1	Sample 2	Sample 3	Sample 1	Sample 2	Sample 3	
**Total raw reads**	54,785,178	55,921,570	59,798,448	57,139,776	56,119,540	53,573,230	337,337,742
**Total clean reads**	54,359,728	55,495,942	59,370,008	56,723,546	55,719,248	53,186,974	334,855,446
**Q20 percentage**	98.82%	98.89%	98.95%	98.84%	98.90%	98.92%	
**GC percentage**	49.42%	49.40%	49.80%	50.13%	48.21%	49.60%	

**Table 3 genes-13-02057-t003:** Summary of assembly statistics and functional annotation of unigenes.

Description	Number
**Assembly statistics**	
Number of transcripts	41,345
Number of unigenes	23,442
**Functional annotation of unigenes**	
GO	3516
KEGG	9826
COG	14,459
NR	15,572
Swiss-Prot	12,814
Pfam	13,113
Total annotation	16,084

## Data Availability

The datasets generated for this study can be found in the NCBI Sequence Read Archive (https://www.ncbi.nlm.nih.gov/sra/?term=PRJNA756609 accessed on 12 October 2022) with reference number PRJNA756609.

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
