# Peer review of "Transcriptome Analysis Reveals That SREBP Modulates a Large Repertoire of Genes Involved in Key Cellular Functions in Penaeus vannamei, although the Majority of the Dysregulated Genes Are Unannotated"

_genes, 2022, doi:10.3390/genes13112057_

Round 1

Reviewer 1 Report

Dear Authors, please find enclosed a few suggestion to your very valuable and interesting manuscript!

With best wishes,

R.

Author Response

Please see the attachment for a detailed point-by-point response to the review comments.

Reviewer 2 Report

This paper describes the role played by SREBP in the Penaeus vannamei shrimp hepatopancreas. A SREBP silencing through RNA interference reveals that this protein is involved in many processes like energy metabolism, immune response and other interesting pathways. I think that the experiments are well organized and the data are interesting. There are some observations I would like the Authors to address:

In the Abstract, row 21, I think that the word “responsive” is referred to the up-regulated genes, but to me every gene which expression changes following a stimulus is responsive, both up- and down-regulated. The phrase is a bit confusing, the authors should correct.

Row 117 the legend of the table is missing

Rows 108-109 I’m sure there are references for the intramuscular injection described that should be added. For example, I found Nguyen et al., 2018, “RNA interference in shrimp and potential applications in aquaculture” for double strand RNA. The Authors should specify the origin of the protocol they applied

Rows 264-272 there are some genes exclusively expressed in one of the two conditions. Usually, in other kind of differential expression studies, those genes would have been analysed. It took a while to me to understand why in this study they are excluded from the subsequent analysis, and I have to say that I agree that this is correct, but I suggest to give a brief discussion of this choice to allow an easier reading of this interesting paper.

Author Response

Kindly see the attachment for a detailed point-by-point response to the review comments.

Reviewer 3 Report

This is an interesting manuscript including RNA interference technique followed by transcriptome analysis wit the objective to explore the genes modulated by SREBP in Penaeus vannamei hepatopancreas. The methodology is described with sufficient detail. Results are clear and concise, and discussion is appropriate and well supported with several references.

I suggest considering next minor comments to improve the manuscript:

1)    Abstract: According to guidelines described in “Instructions for authors”, this section should include a brief description of the main methods or treatments applied.

2)    Introduction: Please briefly mention in the last paragraph the main aim of the work.

3)    Discussion: I suggest highlighting the main conclusions of the work at the end of this section.

4)    References: The list of authors for several references is incomplete. Also, please verify that references 71 and 72 are included in the manuscript.

Minor grammar comments:

1)    Line 19: Replace “are” by “were”.

2)    Line 24: Replace “reveals” by “revealed”.

3)    Line 26: Replace “indicate” by “indicated”.

4)    Line 39: Insert a comma after the round bracket.

5)    Line 56: Remove the word “and” at the middle of the line.

6)    Line 76: Replace “produce” by “produced”.

7)    Line 179: Replace “has been” by “were”.

8)    Line 248: Remove the word “(Table 3)”.

9)    Line 309: Replace “are” by “were”.

10) Line 576: The reference is incomplete.

Author Response

Please see the attachment for a detailed response to the comments.
